# Variations in the oral microbiome and metabolome of methamphetamine users

Dawei Wang,[1] Yu Feng,[1] Min Yang,[2] Haihui Sun,[3] Qingchen Zhang,[4] Rongrong Wang,[5] Shuqing Tong,[6] Rui Su,[6] Yan Jin,[6] Yunshan Wang,[6] Zhiming Lu,[6] Lihui Han,[2] Yundong Sun[7]

**ABSTRACT** Drug addiction can seriously damage human physical and mental health, while detoxification is a long and difficult process. Although studies have reported changes in the oral microbiome of methamphetamine (METH) users, the role that the microbiome plays in the process of drug addiction is still unknown. This study aims to explore the function of the microbiome based on analysis of the variations in the oral microbiome and metabolome of METH users. We performed the 16S rRNA sequencing analysis based on the oral saliva samples collected from 278 METH users and 105 healthy controls (CTL). In addition, the untargeted metabolomic profiling was conducted based on 220 samples. Compared to the CTL group, alpha diversity was reduced in the group of METH users and the relative abundances of *Peptostreptococcus* and *Gemella* were significantly increased, while the relative abundances of *Campylobacter* and *Aggregatibacter* were significantly decreased. Variations were also detected in oral metabolic pathways, including enhanced tryptophan metabolism, lysine biosynthesis, purine metabolism, and steroid biosynthesis. Conversely, the metabolic pathways of porphyrin metabolism, glutathione metabolism, and pentose phosphate were significantly reduced. It was speculated that four key microbial taxa, i.e., *Peptostreptococcus*, *Gemella*, *Campylobacter*, and *Aggregatibacter*, could be involved in the toxicity and addiction mechanisms of METH by affecting the above metabolic pathways. It was found that with the increase of drug use years, the content of tryptamine associated with neuropsychiatric disorders was gradually increased. Our study provides novel insights into exploring the toxic damage and addiction mechanisms underlying the METH addiction.

**IMPORTANCE** It was found that with the increase of drug use years, the content of tryptamine associated with neuropsychiatric disorders gradually increased. The prediction models based on oral microbiome and metabolome could effectively predict the methamphetamine (METH) smoking. Our study provides novel insights into the exploration of the molecular mechanisms regulating the toxic damage and addiction of METH as well as new ideas for early prevention and treatment strategies of METH addiction.

**KEYWORDS** methamphetamine users, detoxification, oral microbiome, oral metabolites, 16S rRNA sequencing

Methamphetamine (METH) is a type of amphetamine-like stimulant that primarily causes addiction and neurotoxic damage to the brain by damaging dopamine and serotonin neurons (1). METH has toxic effects on multiple systems in the body, including the cardiovascular, gastrointestinal, and immune systems (2, 3). At present, drug driving is a serious social problem that affects human safety (4). The long-term use of METH can cause irreparable damage to the brain, while the detoxification is a long

Address correspondence to Yundong Sun, syd@sdu.edu.cn.

Dawei Wang and Yu Feng contributed equally to this article. Author order was determined by drawing straws.

The authors declare no conflict of interest.

See the funding table on p. 15.

and difficult process (5). However, current strategies for the prevention and treatment of METH addiction are limited (6).

It is well known that drug abuse particularly affects the immune system of the human body (3), which is a key factor in the interaction between the microbiome and the host (7). Previous studies have shown that the toxicity damage caused by METH is closely related to the gut microbiome, and could be treated in mouse models by regulating the metabolites derived from the gut microbiome (8). Studies have also found that addiction to METH smoking can disrupt the balance of the human oral microbiome, characterized by decreased microbial diversity, while the oral microbiome is improved during the detoxification process (9). Furthermore, the oral microbiomes are also closely related to systemic diseases such as gastrointestinal and cardiovascular disorders (10).

Microorganisms can transmit signals to the brain through multiple pathways, and the brain in turn regulates the composition of microorganisms through the nervous system (11). Metabolites produced by microorganisms can mediate these signaling pathways and induce host reactions in distant organs (12). Increasing evidence suggests that microbial metabolites, such as tryptamine and isoleucine-leucine, are involved in the molecular mechanisms underlying the toxicity and addiction of METH by affecting amino acid biosynthesis, membrane lipid metabolism, and energy metabolism (13, 14). To date, the oral metabolomics has become a powerful tool to study the association between oral metabolites and diseases (15). However, it is currently unclear whether the metabolites derived from oral microorganisms play a role in the detoxification process of METH addiction. Therefore, integrating data from oral microbiomes and metabolomes could help explore the toxicity and addiction mechanisms of METH and provide novel methods for the recovery of physical and mental health of drug users.

In this study, to investigate the role played by the oral microbiome in METH addiction, we performed the 16S rRNA gene amplicon sequencing and untargeted liquid chromatography–tandem mass spectrometry metabolomic profiling analyses of oral saliva samples from both METH users under detoxification and healthy control (CTL) to investigate the changes in the oral microbiome of METH users and assess the potential functional connections between the oral microbiome and metabolome, providing novel insights into the molecular mechanisms regulating the METH-related toxicity and addiction.

## MATERIALS AND METHODS

### Study population

All participants completed a questionnaire and provided an oral saliva sample. The questionnaire included information of the participant's gender, age, height, weight, body mass index (BMI), and smoking and drinking status. METH-related questions included information of intake method, duration of use, frequency of use, amount of use, and duration of detoxification (16). In our study, smoking was the intake method for METH, and the duration of detoxification was between 6 and 12 months. Subjects were comprehensively examined for oral hygiene according to the oral health criteria established by the World Health Organization: no pain, normal gum color, no bleeding, no caries, and clean teeth. From August 2021 to July 2022, a total of 1,256 saliva samples were collected from Shandong, China, and 383 samples were ultimately included in the subsequent analyses based on the following inclusion criteria: (i) normal oral health status; (ii) normal dietary habit; (iii) no underlying diseases; and (iv) no antibiotic use within the past 4 weeks. Neither the patients nor the CTL were involved in the design, conduct, reporting, or dissemination plans of this research (Table 1; Table S1).

### Saliva sample collection

The subjects were requested to collect saliva samples using a sterile collection tube using the drooling method (9). Briefly, the subjects were instructed not to brush teeth,

**TABLE 1** The characteristics of the study population[a]

| Demographic characteristics | CTL (n = 105) | METH addicts (n = 278) | P-value |
|---|---|---|---|
| Age (year) | 35.97 ± 4.87 | 34.98 ± 5.16 | 0.091 |
| Gender | | | |
| Female | 19 (18.1%) | 45 (16.2%) | 0.656 |
| Male | 86 (81.9%) | 233 (83.8%) | |
| BMI | 24.50 ± 3.77 | 25.54 ± 5.05 | 0.056 |
| Smoking habit | | | |
| Without | 29 (27.6%) | 57 (20.5%) | 0.137 |
| With | 76 (72.4%) | 221 (79.5%) | |
| Alcohol drinking | | | |
| Without | 69 (65.7%) | 186 (66.9%) | 0.826 |
| With | 36 (34.3%) | 92 (33.1%) | |
| Drugs usage | | | |
| Without | 100 (95.2%) | 270 (97.1%) | 0.365 |
| With | 5 (4.8%) | 8 (2.9%) | |
| Oral condition | Normal | Normal | |
| Dietary habit | Mixed diet | Mixed diet | |
| Basic disease | NO | NO | |

[a]For the comparison of all data, Wilcoxon rank-sum test was performed.

drink water, or eat 1 h prior to the collection of saliva samples. Each participant spit the saliva samples into the sterile collection tube through a funnel for three to five times with a 3-min interval, until the liquid portion of the saliva sample (without bubbles) reached the 5 mL graduation line. The final saliva collections should not contain any impurities and phlegm. The samples were immediately stored at −80°C for preservation.

## DNA extraction and 16S rRNA gene amplicon sequencing and data processing

Total DNA was extracted from saliva samples using the TGuide S96 Magnetic Soil/ Stool DNA Kit (Tiangen Biotech Co., Ltd., Beijing, China) following the manufacturer's instructions. The V3–V4 region of the 16S rRNA gene was amplified and sequenced on the Illumina Miseq platform using the universal bacterial primers (341F-59-CCTACGGGGG GGCWGCAG-39 and 806R59-GGACTACHVGGGTWTCTAAT-39) by BioMarker Technologies Co. (Beijing, China) (17) (Fig. 1). The QIIME version 1.9.1 was used to analyze the 16S rRNA sequence data. Sequences with a similarity ≥97% were classified into the same operational taxonomic unit (OTU). Species annotation and classification were performed based on the Silva database (v138.1) (http://www.arb-silva.de/) and RDP Classifier (http://sourceforge.net/projects/rdpclassifier/). Alpha diversity analysis was performed by calculating the ACE, Chao, Shannon, and Simpson indices for each sample. Beta diversity analysis was performed to compare the differences in community composition and structure between different sample groups. Alpha and beta diversity analyses were conducted using q1 diversity. Principal coordinate analysis (PCoA) was used to visualize the separation of the sample groups of METH users and CTL. The significant differences at the genus level between groups were evaluated using Metastats analysis.

## Saliva metabolome profiling and data processing

Saliva metabolite analysis was performed using ultra-high performance liquid chromatography (Waters UPLC Acquity I-Class PLUS, Milford, USA) and high-resolution mass spectrometer (Waters UPLC Xevo G2-XS QTof, Milford, USA). Initially, a total of 100 μL saliva sample was weighed and added with 500 μL extraction solution containing the internal standard (1,000:2; methanol-acetonitrile volume ratio = 1:1; internal standard concentration 2 mg/L). The sample was vortexed for 30 s and then sonicated for 10 min in the ice bath, incubated at −20°C for 1 h, and centrifuged at 12,000 rpm for 15 min at

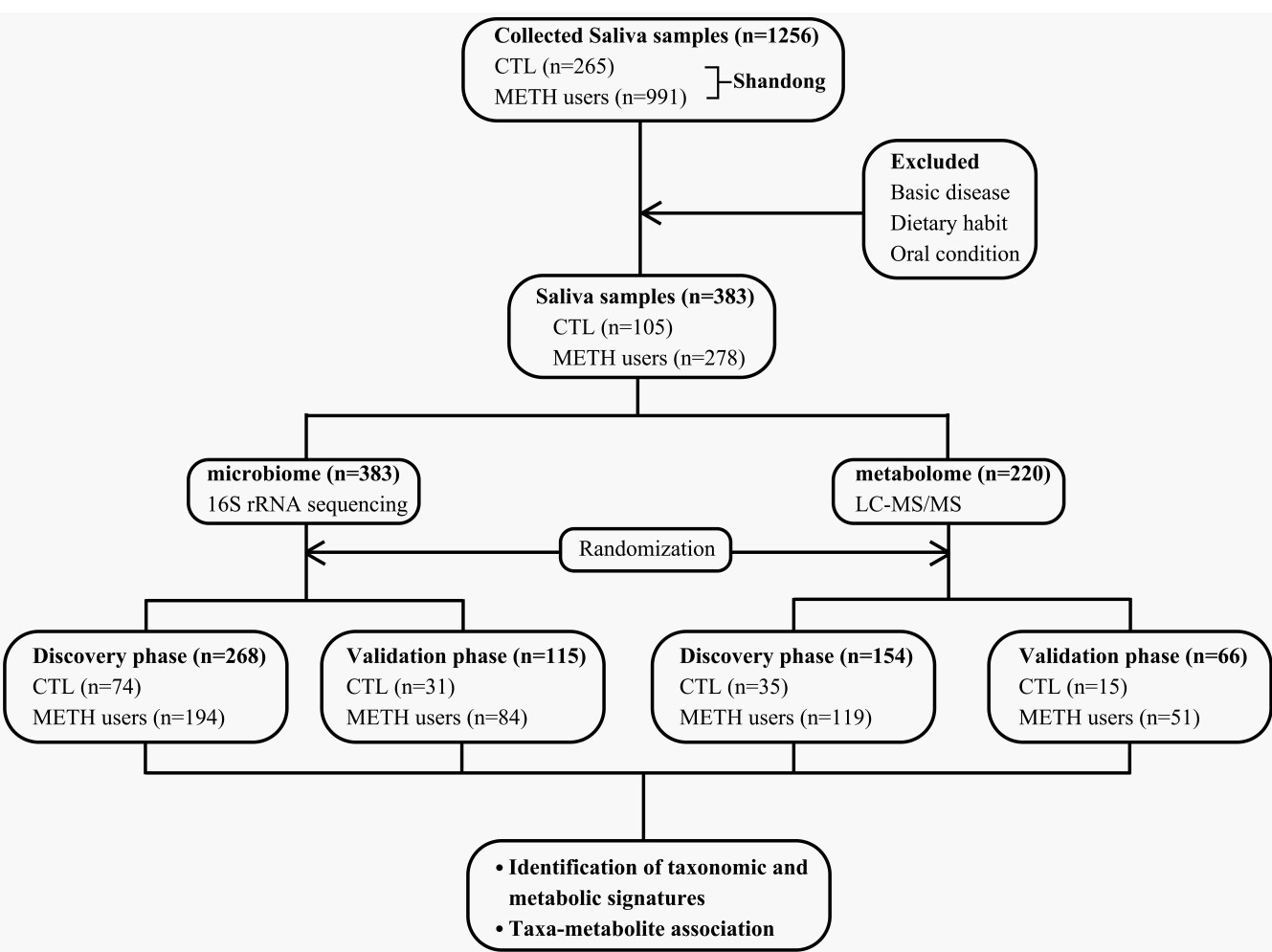

**FIG 1** Study design and flow diagram. A total of 1,256 oral saliva samples are collected from drug rehabs in Shandong, China. Based on strict exclusion criteria, a total of 383 samples are selected for microbial sequencing analysis and 220 samples for metabolomics analysis, with both groups of samples randomly divided into discovery and validation phases. The classifier is constructed based on the random forest model in the discovery phase; the validation phase is used to verify the diagnostic efficiency of the classifier. CTL, healthy control; LCMS/MS, liquid chromatography-tandem mass spectrometry.

4°C. A total of 500 µL supernatant was then removed and dried in a vacuum concentrator. Then, a total of 160 µL extraction solution (acetonitrile and water volume ratio of 1:1) was added to the dried metabolite to repeat the above procedures of vortex, sonication, and centrifugation. Finally, a total of 120 µL supernatant was collected and mixed with 10 µL of each sample to generate the quality control samples.

The first and second mass spectrometry data collections were performed in mass spectrometry (MSe) mode controlled by software MassLynx V4.2 (Waters, Milford, USA). The raw data collected underwent data processing operations such as peak extraction and peak alignment using Progenesis QI software. Metabolite identifications were carried out based on the METLIN database and in-house database of Maybridge using Progenesis QI (18). Theoretical fragment recognition was also performed with the quality deviation within 100 ppm. A total of 2,018 peaks were detected with 229 metabolites annotated and shown in the heat map (Table S2; Fig. S1).

## Statistics

Biochemical data were analyzed and plotted using R language (v4.1.1). Differences in continuous variables and microbiota between two groups were analyzed using the Wilcoxon rank-sum test. The orthogonal partial least squares discriminant analysis

(OPLS-DA) was performed using ropls (1.6.2) for comparison between two groups. The log2 transformed heat maps of microbiota were plotted based on the 0 value of 1E-05. The metabolite heat map was plotted after log10 conversion (to prevent expressions with a quantification of 0, the value of each quantification result was added by 1). Spearman rank correlation test (Presiduals Package) was performed to assess the association between key microbiota and key metabolites.

Random forest analysis was performed using the R language (v3.6.1) randomForest package (v4.6–14) and visualized using the pROC package (v1.18.0) and the ggplot2 package (v3.3.3). The receiver operating characteristic (ROC) effects were represented using the area under the ROC curve (AUC). For the input features, the classifiers contained only key microbiota and metabolites. According to the previous research experience, Bootstrap sampling method was chosen as the sampling method, with both strata and sampsize options to overcome the issue of sample imbalance (19). Additionally, the number of decision trees was set as 500 (20).

## RESULTS

### Study design and flow

A total of 1,256 oral saliva samples were collected from CTL and METH users undergoing detoxification in a detoxification facility in Shandong, China. Based on the strict exclusion criteria, a total of 383 participants was chosen for further analysis, including the group of METH users ($n = 278$) and the CTL group ($n = 108$). The microbial sequencing was performed on all samples, and 220 samples (170 METH users and 50 CTL) were subjected to metabolomics; in both the microbial sequencing and metabolomics analysis, samples were randomly divided into discovery and validation phases (Fig. 1). In the discovery phase (194 METH users and 74 CTL) of microbial sequencing, the key microbial markers were identified and the classifiers were generated using the random forest model; in the validation phase (84 METH users and 31 CTL), the diagnostic efficiency of the classifier was verified. In the discovery phase of metabolomics (119 METH users and 35 CTL), the key metabolite markers were identified and the classifiers were generated based on the random forest models; in the validation phase, the diagnostic efficiency of the classifiers was verified based on a total of 51 METH users and 15 CTL.

### Characterization of the oral microbiota and oral metabolome of METH users

Sparse analysis based on the discovery cohort of the 16S rRNA sequences showed that the estimated OTU richness was essentially close to saturation in each group (Fig. S2). The alpha diversity was significantly lower in the group of METH users ($n = 194$) compared to the CTL group ($n = 74$) (Fig. 2A). The PCoA was performed based on binary jaccard distance, showing significant differences in community composition of the oral microbiota between the groups of CTL and METH users (Fig. 2B). The untargeted metabolomic profiling of the saliva samples was performed to evaluate the interaction between oral microbiota and host metabolism. The OPLS-DA scores showed that the metabolite compositions of the groups of METH users ($n = 119$) and CTL ($n = 35$) were dispersed in two different regions (R2Y = 0.533 and Q2Y = 0.404; $P < 0.05$) (Fig. 2C). The functional pathways involved in the metabolites were further analyzed using the Kyoto Encyclopedia of Genes and Genomes (KEGG) database. The results showed that metabolic pathways of tryptophan metabolism, lysine biosynthesis, purine metabolism, and steroid biosynthesis were enhanced in the group of METH users, while the metabolic pathways of porphyrin metabolism, glutathione metabolism, and pentose phosphate were significantly decreased (Fig. 2D). The oral microbial compositions of each sample in the two groups were comparatively analyzed at the phylum and genus levels (Fig. 2E and F).

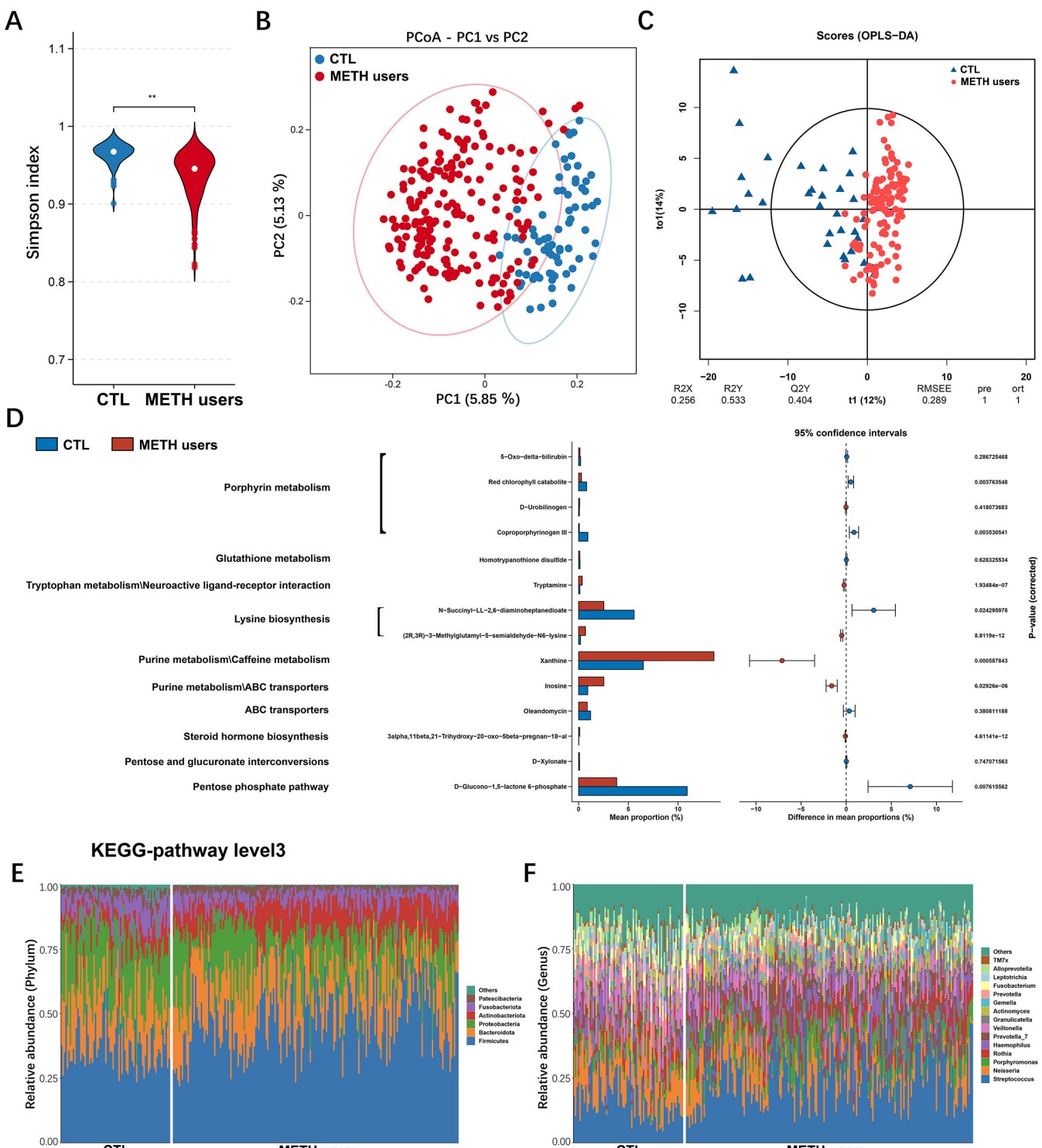

**FIG 2** Characteristics of oral microbiota and metabolome of the groups of CTL and METH users. (A) Alpha diversity analysis. (B) PCoA based on binary jaccard distance showing the different classification compositions between the groups of METH users and CTL. (C) Oral salivary metabolome profiles of the groups of CTL and METH users based on the OPLS-DA. (D) Selection of different gene categories according to significant differences in level 3 gene categories (*P* < 0.05). (E) Composition of the oral microbiota at the phylum level for all samples. (F) Composition of the oral microbiota at the genus level for all samples. Each experiment is repeated three times. **P* < 0.01.

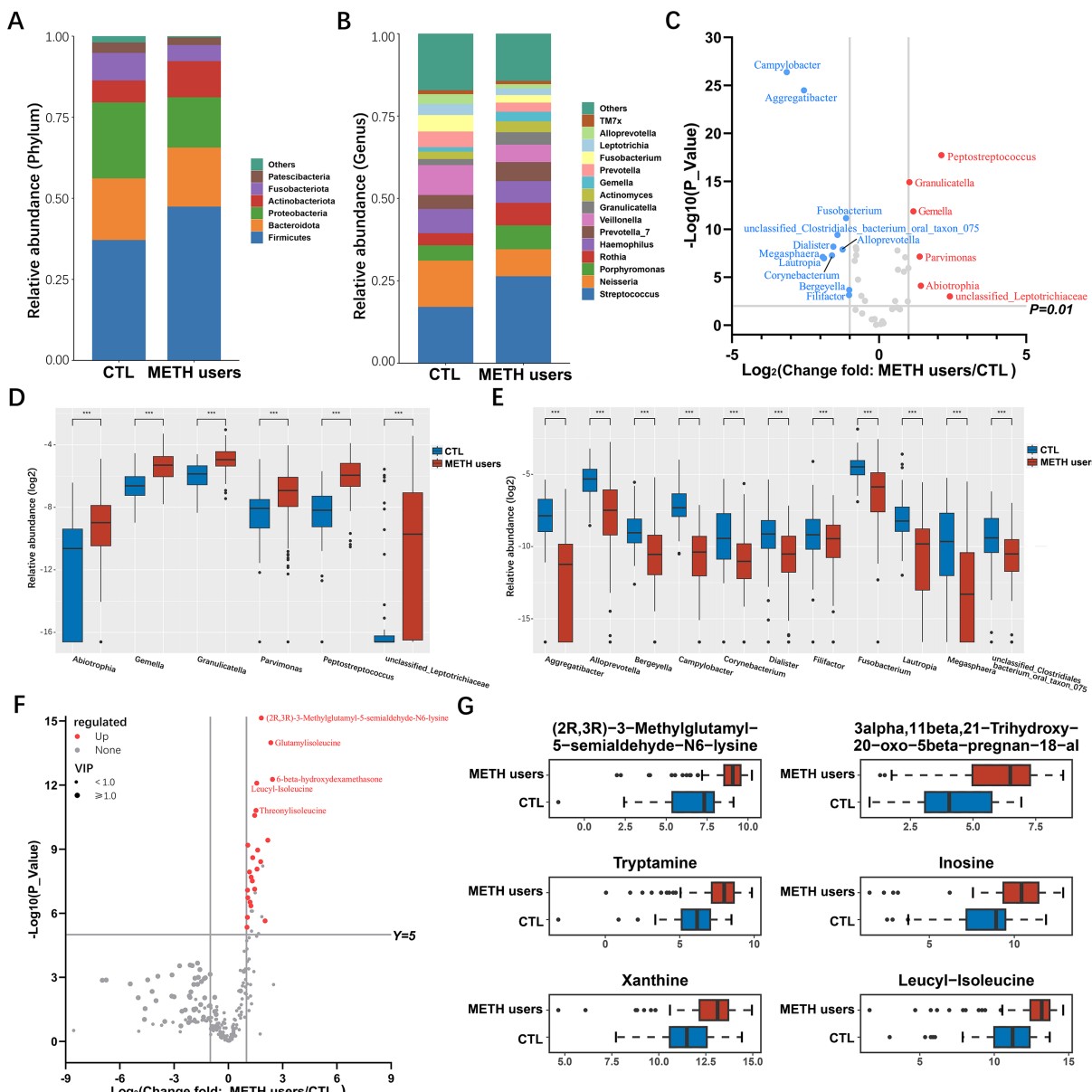

**FIG 3** Variations in the oral microbiota and metabolome in the groups of CTL and METH users. (A) Comparative analysis of the oral microorganisms between the groups of CTL and METH users at the phylum level showing microbial taxa with the mean relative abundance >1%. (B) Comparative analysis of the oral microorganisms between the groups of CTL and METH users at the genus level showing microbial taxa with a mean relative abundance >1%. (C) Volcano plot showing the changes in oral microbiota in the groups of CTL and METH users ($P < 0.01$) based on bacteria with relative abundance >0.1% at the genus level ($n$ = 17). (D) Increased and (E) decreased key microbial communities at the genus level in the group of METH users compared to the CTL group. (F) Volcano plot analysis of changes in oral salivary metabolite grouping in the groups of CTL and METH users ($P < 0.001$). Elevated metabolites are highlighted in red ($n$ = 23) and the top 5 metabolites with the lowest $P$-values are marked with text. (G) Six representative key metabolite classes are selected based on the significant differences in (F). Relative abundances were logarithmic-transformed and 0 values were assigned 1e-05. Each experiment is repeated three times. Data are presented as mean ± standard deviation ($n$ = 3). ***$P < 0.001$.

## Changes in the oral microbiota of METH users

The compositions of the oral microbiota between the groups of METH users and CTL were comparatively evaluated at the phylum and genus levels (Fig. 3A and B), showing the microbial taxa with the average relative abundance >1%. At the phylum level, the three dominant populations in the two groups of oral microbiota (i.e., Bacteroidota, Firmicutes, and Proteobacteria) accounted for an average of more than 80% of the

sequences. At the genus level, *Streptococcus*, *Neisseria*, *Porphyromonas*, *Rothia*, *Haemophilus*, and *Prevotella_7* microorganisms were identified as the major taxa in both groups of oral microbiota. The volcano map showed the bacteria (*n* = 17) with relative abundance greater than 0.1% and significant difference at the genus level (Fig. 3C). The relative abundances of six microbial genera (e.g., *Granulicatella*, *Gemella*, and *Peptostreptococcus*) were significantly increased in the group of METH users compared to the CTL group (Fig. 3D), while the relative abundances of 11 genera (e.g., *Campylobacter*, *Aggregatibacter*, and *Dialister*) were significantly reduced (Fig. 3E). The heat map was generated to show the relative abundances of key bacteria in each sample (Fig. S3).

## Changes in the oral metabolome of METH users

The volcano plots showed that the relative abundances of 23 metabolites in the group of METH users were significantly increased in comparison with the CTL group (Fig. 3F). The significantly different metabolites were identified based on the following three conditions: (i) variable importance in the projection of (VIP) ≥1, (ii) fold change (FC) >2 or <0.5, and (iii) −log10 (*P*-value) >5 (Table S3). A total of six representative key metabolites were selected and presented in box line plots (Fig. 3G), including tryptamine, leucyl-isoleucine, inosine, xanthine, (2R,3R)-3-methylglutamyl-5-semialde-N6-lysine, and 3alpha,11beta,21-Trihydroxy-20-oxo-5beta-pregnan-18-al.

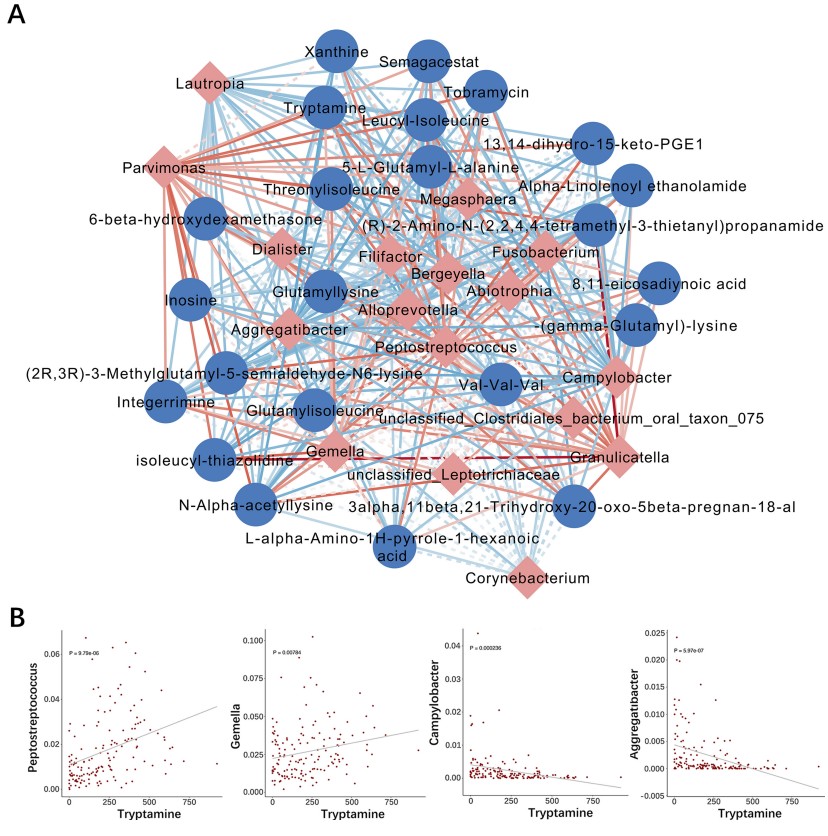

**FIG 4**  An integrated association network of microbiota-metabolite interactions. (A) Network showing the correlations between key microorganisms (*n* = 17) and key metabolites (*n* = 23) with significant difference in the group of METH users based on Spearman analysis (*P* < 0.05). The red nodes represent microorganisms and the blue nodes represent metabolites. The lines connecting the nodes indicate either positive (red) or negative (blue) correlations. Solid and dashed lines indicate significant and implied correlations, respectively. (B) Examples of microbial taxa correlated with metabolites with each point representing a sample.

## Association between oral microbiota and metabolites in METH users

The correlation analysis (Spearman analysis, $P < 0.05$) between oral metabolites ($n = 23$) and oral microorganisms ($n = 17$) that differed significantly in the group of METH users ($n = 119$) was performed to explore the association between oral microbiota and metabolites (Fig. S4). The results confirmed the close association between oral microorganisms and oral metabolites. The interaction between differed significantly microorganisms ($n = 17$) and metabolites ($n = 23$) was further evaluated using an integrated association network (Spearman analysis; $P < 0.05$) (Fig. 4A). The results showed that tryptamine was positively correlated with *Gemella* and *Peptostreptococcus* and negatively correlated with *Campylobacter* and *Aggregatibacter* (Fig. 4B).

The top 50 microorganisms and metabolites with the highest abundances were screened based on the Spearman's correlation coefficient ≥0.1 and $q ≤ 0.05$ as filtering parameters to further investigate the correlation between oral microbiota and metabolites (Fig. 5). The results of the top 10 bacteria or metabolites revealed strong interactions between the two groups of oral microorganisms or metabolites. In the microorganisms, the group of METH users showed more positive interactions between oral microorganisms compared to the CTL group (Fig. 5A and B). For the metabolites, the group of METH users showed increased levels of oral metabolite interactions compared to the CTL group, exhibiting a strong positive correlation profile (Fig. 5C and D).

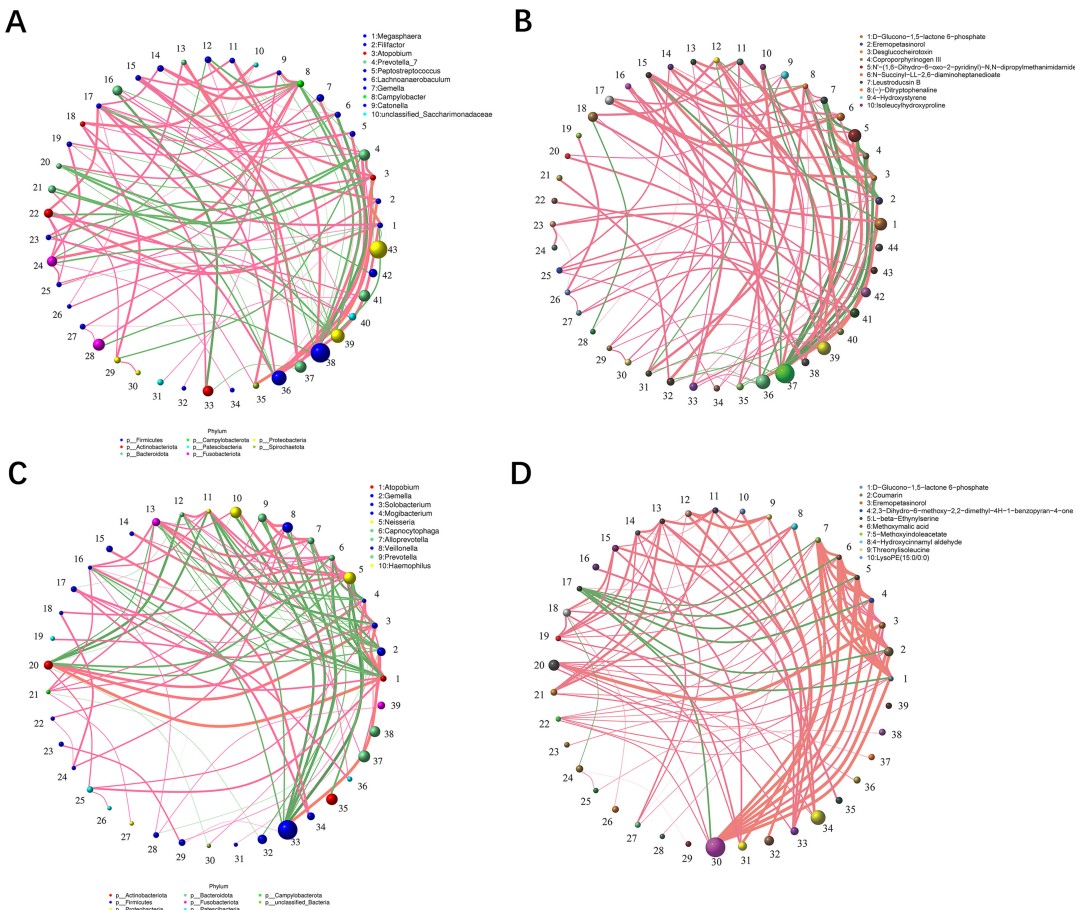

**FIG 5** The respective interactions between the two groups of oral microorganisms and metabolites. (A) Interaction between the levels of oral microbial genera in the CTL group. (B) Interaction relationships between oral metabolites in the CTL group. (C) Interaction between the levels of oral microbial genera in the group of METH users. (D) Interaction relationship between oral metabolites of the group of METH users. The size of the circle represents the relative abundance, the line represents the correlation between the two taxa at both ends of the line, the thickness of the line represents the strength of the correlation, the orange line represents a positive correlation, and the green line represents a negative correlation.

## Characterization of oral microbiota and oral metabolome in METH users with different years of drug use

To further analyze the effect of duration of drug use on the oral microbiota and metabolome, the METH users were regrouped according to their years of drug use (0–2 years, 3 years, 4–5 years, 6–9 years, and ≥10 years) (Table S4). The alpha diversity analysis showed significant differences in the oral microbiota between the groups of CTL and METH users (Fig. S5). The results of PCoA analysis showed that the spatial distributions of oral microbiota in the group of METH users were varied significantly based on the different durations of drug use (Fig. 6A). The OPLS-DA scores showed that the metabolic compositions of the group of METH users were scattered in different regions from the CTL group (R2Y = 0.74 and Q2Y = 0.492; $P < 0.05$), and the differences were more significant with the increasing years of drug use (Fig. 6B). Among them, the amount of tryptamine gradually increases with the increase of drug use years (Fig. 6C). The results of the comparative analysis of oral microbiota between the groups of CTL and METH users performed at the phylum and genus levels (Fig. 6D and E) revealed the microbial taxa with the average relative abundance >1%, showing that the compositions of the oral microbiota were changed with the increasing years of drug use. For example, at the phylum level, the proportion of Firmicutes was gradually increased and the proportion of Bacteroidota was gradually decreased (Fig. S6). At the genus level, the proportions of *Streptococcus* and *Rothia* were gradually increased and those of *Neisseria* were gradually decreased. The oral microbial compositions of each sample in both groups of CTL and METH users were compared at the phylum and genus levels (Fig. S7).

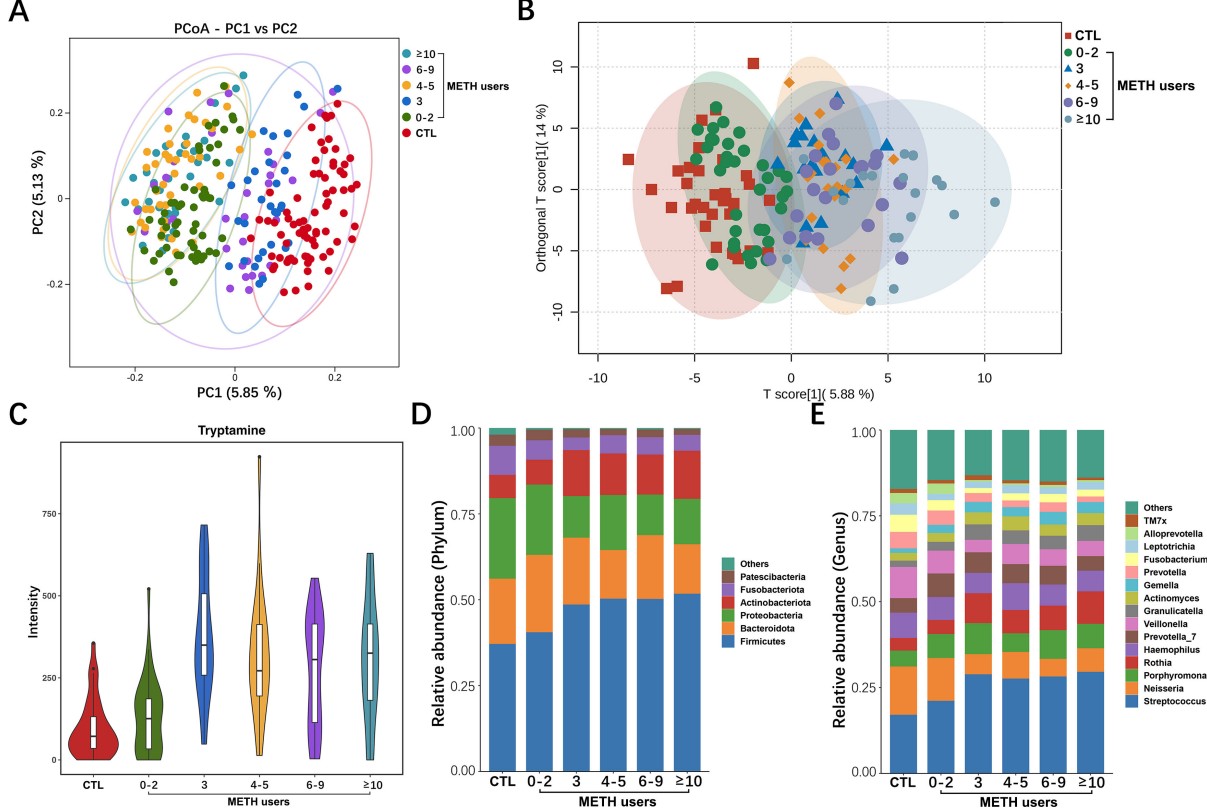

**FIG 6** Characterization of the oral microbiota and metabolomic features in six groups of samples. (A) PCoA based on binary jaccard distance showing the different classification compositions between the groups of CTL and METH users. (B) Oral salivary metabolome profiles in six groups of samples based on OPLS-DA. (C) The violin plot displays the intensity changes of the metabolite tryptamine in six groups. (D) Comparative analysis of oral microbial differences between groups performed at the phylum level showing the microbial taxa with a mean relative abundance >1%. (E) Comparative analysis of oral microbial differences between groups performed at genus level showing the microbial taxa with the mean relative abundance >1%. Each experiment is repeated three times. Data are presented as mean ± standard deviation (*n* = 3).

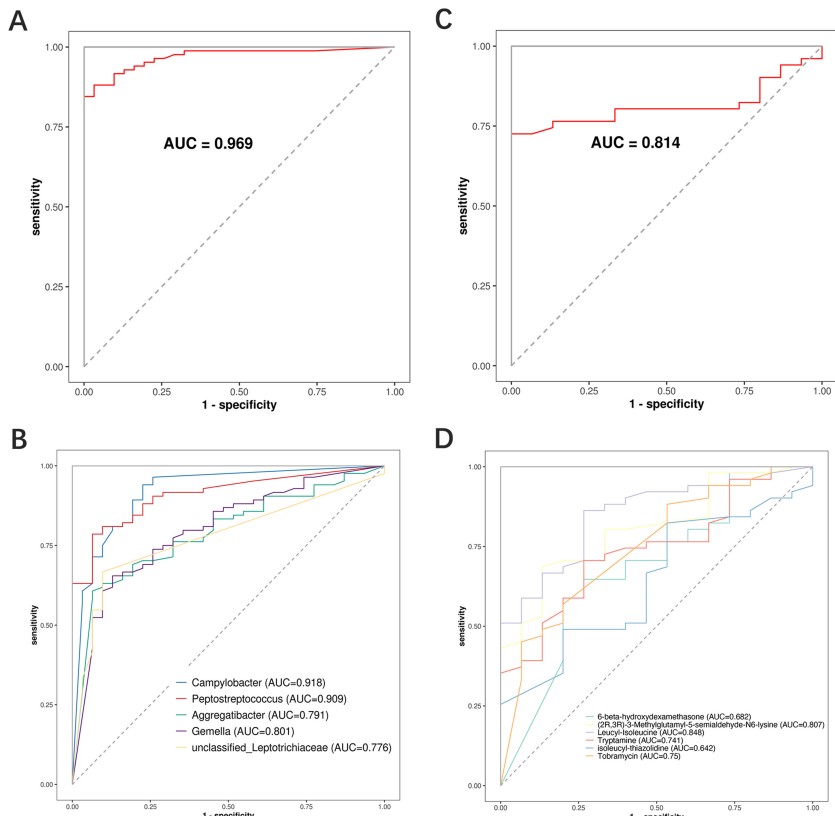

**FIG 7** Identification of biomarkers associated with METH addiction based on the random forest models. (A, B) Random forest classifiers constructed using microorganisms at the genus level (*n* = 5). (C, D) Random forest classifiers constructed based on metabolites (*n* = 6).

## Identification and validation of oral microbial and metabolite biomarkers in METH users

A random forest model was established to investigate the potential application of oral microorganisms and metabolites in identifying the smoking of METH (Table S5). The results showed an AUC value of 0.969 based on the model of five microbial taxa and 0.814 based on the model of six metabolites (Fig. 7). These results were validified by the random forest model re-established based on the top 2 microorganisms and metabolites, respectively. The results showed that the AUC values based on the 2-microorganism and 2-metabolite models were 0.976 and 0.83, respectively (Fig. 8). The AUC value of 0.96 was achieved based on the model of combined two microorganisms and two metabolites (Fig. 8), without significantly improving the accuracy of classification. These data suggested that the prediction of METH smoking based on oral microbiome performed better than the oral metabolites.

The random forest models were re-established based on the number of years of drug use among the METH users to validate the reliability of microbial biomarkers (Fig. S8). The results showed that the AUC values based on the five-microbe model of different durations of drug use were 0.905, 0.988, 1.000, 1.000, and 0.998, respectively (Table S6). The AUC values based on the TOP2 model of microorganisms were 0.925, 0.987, 0.996, 0.993, and 0.995, respectively (Table S6). These data suggested that the five microbial biomarkers were also highly accurate in predicting the status of METH smoking in persons with different durations of drug use.

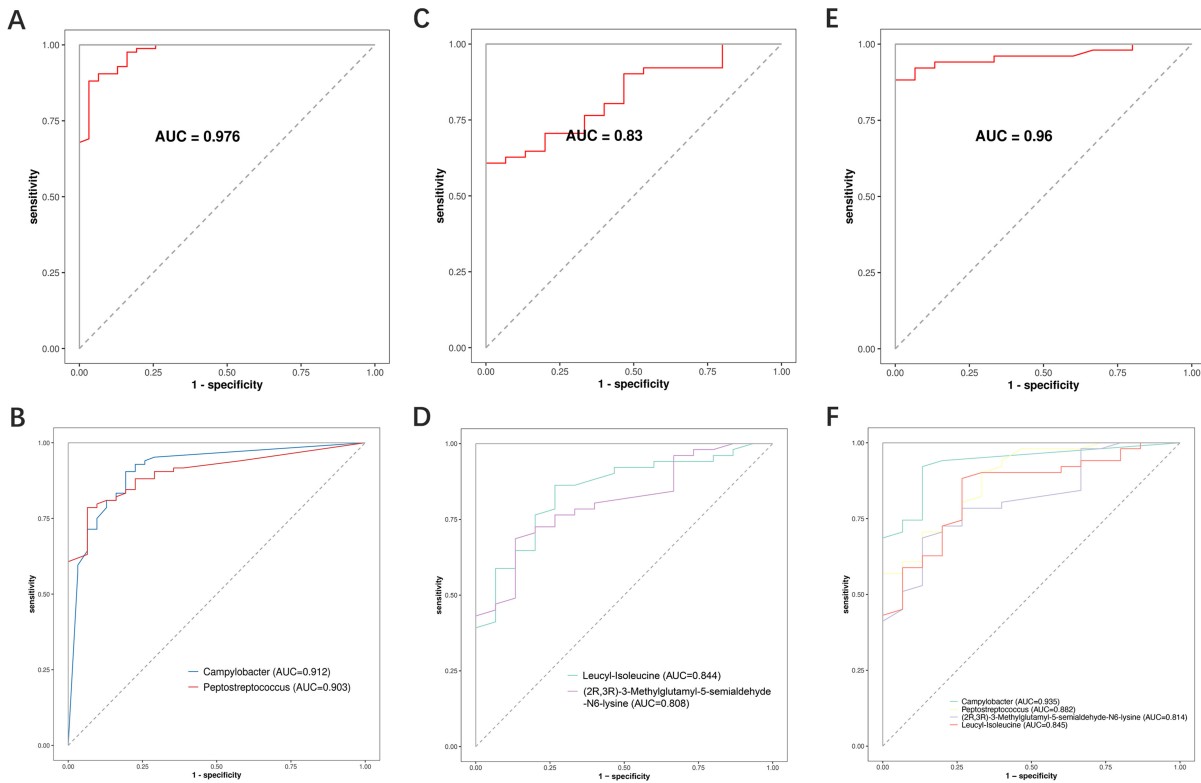

FIG 8   Identification of biomarkers associated with METH addiction based on the random forest models. (A, B) Random forest classifiers constructed using the microorganisms at the genus level (*n* = 2). (C, D) Random forest classifiers constructed based on metabolites (*n* = 2). (E, F) Random forest classifiers constructed using jointly the microorganisms at the genus level (*n* = 2) and metabolites (*n* = 2).

## Validation of oral biomarkers in METH users from different geographical areas

The random forest model was re-established based on the geographical origin of the samples to validate the reliability of the microbial biomarkers (Fig. S9). In the discovery phase, the classifiers based on random forest model were generated with samples from Zibo as the control. In the validation phase, samples from Qingdao and Jinan were used to verify the diagnostic efficiency of the classifiers. The results showed that the AUC value based on the five-microbe model in the validation phase of the Jinan samples was 0.85 (Table S7). The AUC value based on the TOP2 model among microorganisms was 0.83 (Table S7). In the validation phase, the AUC value based on five-microbe model of the Qingdao samples was 0.925 (Table S7). The AUC value based on the TOP2 model in microorganisms was 0.86 (Table S7). These data suggested that the microbial markers were accurate in predicting the status of METH smoking across the geographic regions.

## DISCUSSION

In this study, we performed the combined multi-omics analyses of the oral microbiome and metabolome of METH users under detoxification. Our results showed that even after 6 months of detoxification from METH, the oral microbiome and metabolome of these individuals were significantly different from those of healthy normal subjects, and these differences were characterized by molecular mechanisms underlying the toxic impairment and addiction of METH. Furthermore, our study identified a group of microbial markers associated with METH addiction, showing significant potential for accurate and effective detection of drugged driving.

METH enhances dopamine and norepinephrine levels in the synaptic gap, and long-term use of METH causes neurotoxicity in axon terminals (21). These toxic effects

can lead to not only neuropsychiatric disorders such as stroke and Parkinson's (22), but also adverse effects on the cerebrointestinal pumping and immune system (3). The cardiovascular diseases are the leading cause of death in METH users, especially after overdose (23). However, there are still insufficient studies on the molecular mechanisms of toxic damage and addiction of METH, leading to the inadequate treatment of METH detoxification based on psychotherapy with no specific medication available (6).

## Variations in the oral microbiome

The oral cavity provides a highly heterogeneous ecological niche for microorganisms, and the damage to the oral microbial community caused by drug addiction is generally not permanent (9). In our study, as one of the dominant phyla of oral microbiota observed in METH users, Firmicutes were revealed with significant changes of reduced alpha diversity, microbial community shifts, and alterations in the abundance of individual taxa. At the genus level, the alterations were observed in key microbiota probably involved in the mechanisms underlying the METH toxic injury. In the group of METH users, both *Peptostreptococcus* and *Gemella* were significantly increased, while both *Campylobacter* and *Aggregatibacter* were significantly decreased, compared to the CTL group.

Studies have shown that *Peptostreptococcus* was positively correlated with the cardiometabolic biomarker high-sensitivity C-reactive protein, which was involved in the cardiovascular diseases (24, 24). Furthermore, as a potential mediator of microbiome ecological dysregulation, *Peptostreptococcus* is enriched in the gut microbiota of patients with autoimmune diseases and ischemic stroke (25, 26). *Gemella* is also a part of the human oral microbiome. As the most common infection caused by *Gemella*, the infective endocarditis is also positively associated with the cardiovascular outcomes (27). Furthermore, *Gemella* was also enriched in the oral microbiome of Parkinson's patients and smokers and was positively correlated with pro-inflammatory cytokines (28). Studies have shown that catecholamines, norepinephrine, and dopamine promote the growth of *Campylobacter* in a strain-dependent manner and its pathogenicity (29). It has been reported that *Aggregatibacter* may promote the development of neurological diseases with the *Aggregatibacter* aggregates detected in patients with Parkinson's disease (30).

Recently, Zhang et al. found that METH addiction caused a decrease in alpha diversity of the oral microbiota and an increase in alpha diversity during detoxification with Bacteroidota identified as the dominant phylum and a group of five key microorganisms (i.e., *Neisseria subflava*, *Haemophilus parainfluenzae*, *Fusobacterium periodonticum*, *Prevotella melaninogenica*, and *Veillonella dispar*) involved in influencing the mechanisms regulating the METH addiction, providing high predictive accuracy for differentiating METH users based on a random forest classifier (9). However, our study revealed different variations in oral microbiota, probably due to (i) the varied geographical sources of the samples and (ii) the length of detoxification, the smoking status, and the oral health of the participants, which could affect the compositions of the oral microbiota (31). Considering that different microorganisms could perform similar functions (32), the molecular mechanisms underlying the toxic damage and addiction of METH cannot be fully explained from the microbial perspective alone.

## Variations in the saliva metabolome

To date, the growing evidence suggests that microorganisms perform different functions by the metabolites they produce, and the metabolites have become the important bridges between microorganisms and diseases in microbial pathogenesis (33). Integrated analysis of multi-omics from microbiome and metabolome can provide clues to the mechanistic connections between microbiome and diseases (34). Therefore, we performed a metabolomic analysis of the saliva samples. Our results showed that many differentially expressed metabolites in the group of METH users were associated with the metabolisms of amino acids, nucleotides, lipids, carbohydrates, cofactors, and vitamins. In particular, the metabolic pathways of tryptophan metabolism, lysine biosynthesis,

purine metabolism, and steroid biosynthesis were enhanced in the group of METH users, while the metabolic pathways of porphyrin metabolism, glutathione metabolism, and the pentose phosphate were significantly reduced.

Currently, the tryptophan metabolic pathway is considered a major pathway connecting multiple systems and is closely associated with neuropsychiatric disorders (35). It was found that 5-HT2A receptors were the key mediators of the addictive effects of METH, and tryptamine could inhibit the basal electrical activity of dopamine neurons by participating in tryptophan metabolism to produce 5-hydroxytryptamine (5-HT) (36). Furthermore, METH has been reported to interact with the re-uptake of 5-HT, resulting in an acute increase in extracellular level of 5-HT, which, in addition to dopamine, is a key mechanism underlying the drug addiction (37). Moreover, a recent study reported that the withdrawal syndrome following the termination of drug addiction is also associated with profound changes in 5-HT activity (38). Leucyl-isoleucine is involved in leucine biosynthesis, and mutations in leucine-rich kinases are the common causes of Parkinson's disease, leading to dopamine neuron loss and motor dysfunction through abnormal increases in neuronal protein synthesis (39). The (2R,3R)-3-methylglutamyl-5-semialde-N6-lysine is an amino acid amide involved in lysine biosynthesis. Previous studies showed that trimethylation of lysine was enhanced in the voxel nuclei of the brain in a METH-induced behavioral sensitization model (40). Furthermore, acetylation of lysine plays an important regulatory role in cardiovascular disease (41). Both xanthine and inosine are involved in purine metabolism, while both purine and pyrimidine-related metabolites (e.g., xanthine base and adenosine 5′-monophosphate) are sensitive to METH addiction (42). Studies have shown that purine metabolism is also closely associated with cardiovascular disease, metabolic syndrome, and chronic kidney disease (43). The 3alpha,11beta,21-Trihydroxy-20-oxo-5beta-pregnan-18-al is involved in steroid hormone biosynthesis, while corticosterone exposure enhances METH-induced vascular damage, neuroinflammation, neurodegeneration, and lethality (44). It has been reported that steroid hormones are involved in the pathogenesis of both Parkinson's disease and cardiovascular disease and are associated with the activation of pro-inflammatory mechanisms (45). Glutathione helps maintain the normal immune system function, showing antioxidant effects to protect neurons from oxidative damage (46). In addition, both the porphyrin metabolism and activation of the pentose phosphate pathway are associated with neuroprotection (47, 48).

Based on the integrated multi-omics data, our study revealed strong associations between oral microbiota and metabolites in METH users. In particular, four key microorganisms, i.e., *Peptostreptococcus*, *Gemella*, *Campylobacter*, and *Aggregatibacter*, were significantly associated with six key metabolites, including tryptamine, leucyl-isoleucine, (2R,3R)-3-methylglutamyl-5-semialde-N6-lysine, 3alpha,11beta,21-Trihydroxy-20-oxo-5beta-pregnan-18-al, xanthine, and inosine (Fig. 4A). These results were consistent with those previously reported, i.e., *Peptostreptococcus*, *Gemella*, and *Campylobacter* were involved in the pathogenesis of diabetes, depression, and oral cancer via the tryptophan metabolic pathway, respectively (49–51). Furthermore, the association between *Peptostreptococcus* and steroid hormone synthesis as well as purine metabolism has been detected in gastric cancer and premature adrenal disease (52, 53). Therefore, it was hypothesized that these four key microorganisms (i.e., *Peptostreptococcus*, *Gemella*, *Campylobacter*, and *Aggregatibacter*) are involved in the molecular mechanisms regulating the toxic damage and addiction of METH via the metabolic pathways of tryptophan metabolism, lysine biosynthesis, purine metabolism, and steroid biosynthesis. In conclusion, our study demonstrated that METH disrupted the oral microbial ecological balance and it was difficult to recover the oral microbiota even after 6 months of detoxification, while the alterations in the compositions of oral microbiota and metabolites could be involved in the toxic damage and addiction process of METH.

The limitations of this study are noted. First, the female samples were underrepresented in the METH users. Second, most of the participants were smokers, while only the dietary habits of the participants were obtained through the questionnaire, potentially

confounding the results of this study by the lifestyle and dietary characteristics of the participants. Finally, the lack of blood samples from participants in our study prevented further analysis of oral microbiota and metabolites with clinical indicators. Although our study revealed a functional connection between microbiome and metabolome, it could not define a cause and effect relationship. Further studies based on a more comprehensive sampling, i.e., with blood samples and dietary characteristics on the oral microbiome and metabolome, are needed to verify the findings revealed in our study.

In this study, we provided novel insights to explore the molecular mechanisms underlying the toxic impairment and addiction associated with METH using multi-omics analyses. For the first time, we characterized the variations in the oral metabolome of METH users and identified the potential functional connections between the oral microbiome and metabolome of METH users.

## ACKNOWLEDGMENTS

We would like to thank BMK Biotechnology Co., Ltd. (http://www.biomarker.com.cn/) and its technical platform BMKCloud (www.biocloud.net) for analysis support in 16S rRNA gene sequencing and LC-MS/MS-based metabolomics researches.

This study was financially supported by the National Natural Science Foundation of China (grant/award numbers: 81972057 and 82172313) and the Major Innovation Project of Shandong Province (grant/award number: 2021GXGC011305).

Study concept and design: Y.S. and D.W. Acquisition of data: R.W., S.T., and R.S. Analysis and interpretation of data: Y.F., M.Y., H.S., and Q.Z. Technical and material support: D.W., Y.J., Y.W., Z.L., and L.H. Drafting of the manuscript: Y.S. and Y.F. All authors contributed to the article and approved the submitted version.

## AUTHOR AFFILIATIONS

[1]Department of Orthopedic, Shandong Provincial Hospital Affiliated to Shandong First Medical University, Jinan, Shandong, China
[2]Department of Immunology, Shandong Provincial Key Laboratory of Infection and Immunology, School of Basic Medical Sciences, Cheeloo College of Medicine, Shandong University, Jinan, Shandong, China
[3]Department of Cardiology, Shandong Provincial Hospital Affiliated to Shandong First Medical University, Jinan, Shandong, China
[4]Department of Orthopedics, Central hospital affiliated to Shandong First Medical University, Jinan, Shandong, China
[5]Department of Clinical Laboratory, Shandong Provincial Hospital, Shandong University, Jinan, Shandong, China
[6]Department of Clinical Laboratory, Shandong Provincial Hospital Affiliated to Shandong First Medical University, Jinan, Shandong, China
[7]Department of Microbiology, Key Laboratory for Experimental Teratology of Ministry of Education, School of Basic Medicine, Cheeloo College of Medicine, Shandong University, Jinan, Shandong, China

## AUTHOR ORCIDs

Yu Feng  http://orcid.org/0000-0002-1362-5048
Zhiming Lu  http://orcid.org/0000-0003-1228-5739
Yundong Sun  http://orcid.org/0000-0001-5260-2212

## FUNDING

| Funder | Grant(s) | Author(s) |
| --- | --- | --- |
| MOST | National Natural Science Foundation of China (NSFC) | 81972057 | Dawei Wang |

| Funder | Grant(s) | Author(s) |
|---|---|---|
| MOST \| National Natural Science Foundation of China (NSFC) | 82172313 | Yundong Sun |
| Major innovation project of shandong province | 2021GXGC011305 | Yundong Sun |

## AUTHOR CONTRIBUTIONS

Dawei Wang, Data curation, Funding acquisition, Resources, Writing – review and editing | Yu Feng, Formal analysis, Software, Writing – original draft | Min Yang, Formal analysis, Investigation, Project administration | Haihui Sun, Conceptualization, Methodology, Resources | Qingchen Zhang, Investigation, Resources, Software | Rongrong Wang, Investigation, Methodology, Project administration | Shuqing Tong, Data curation, Investigation, Project administration | Rui Su, Methodology, Resources, Software | Yan Jin, Resources, Software, Supervision | Yunshan Wang, Investigation, Methodology, Project administration | Zhiming Lu, Resources, Software, Supervision | Yundong Sun, Formal analysis, Funding acquisition, Project administration, Writing – review and editing.

## DATA AVAILABILITY

The 16S rDNA data sets generated and/or analyzed during the current study are available in the (NCBI) repository, (https://www.ncbi.nlm.nih.gov/sra/) with the accession number PRJNA970411. The untargeted metabolomic profiling data sets generated and/or analyzed during the current study are available in the (MTBLS) repository, (www.ebi.ac.uk/metabolights/) with the accession number MTBLS7807.

## ETHICS APPROVAL

This study was conducted in accordance with the guidelines of the World Medical Association and the Helsinki Declaration.

Ethical approval for this study was granted by the Institutional Review Board of Shandong Provincial Hospital Affiliated to Shandong First Medical University (SWYX: NO. 2023-167), and all participants provided written informed consent.

## ADDITIONAL FILES

The following material is available online.

### Supplemental Material

**Supplemental figures (mSystems00991-23-S0001.docx).** Fig. S1-S9.
**Legends (mSystems00991-23-S0002.docx).** Legends for Tables S1 to S7 and Fig. S1 to S9.
**Supplemental tables (mSystems00991-23-S0003.xlsx).** Tables S1 to S7.

### Open Peer Review

**PEER REVIEW HISTORY (review-history.pdf).** An accounting of the reviewer comments and feedback.

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
