## [Reviewer comments · mSystems]

Variations in the oral microbiome and metabolome of methamphetamine users

Dawei Wang, Yu Feng, Min Yang, Haihui Sun, Qingchen Zhang, Rongrong Wang, Shuqing Tong, Rui Su, Yan Jin, Yunshan Wang, Zhiming Lu, Lihui Han, and Yundong Sun

Corresponding Author(s): Yundong Sun, Shandong University

Review Timeline:

Submission Date:	September 15, 2023
Editorial Decision:	November 2, 2023
Revision Received:	November 7, 2023
Accepted:	November 9, 2023

Editor: Christopher Marshall

Reviewer(s): The reviewers have opted to remain anonymous.

Transaction Report:

DOI: <https://doi.org/10.1128/msystems.00991-23>

Re: mSystems00991-23 (Alterations in the oral microbiome and metabolome of methamphetamine users)

Dear Prof. Yundong Sun:

Revision Guidelines

Sincerely,
Christopher Marshall
Editor
mSystems

Reviewer #1 (Comments for the Author):

The authors have investigated the functions (e.g., detoxification) of oral microbiome of methamphetamine (METH) users using various well-established methodologies, including 16S rRNA sequencing and metabolomic profiling. The main discoveries included that the alpha diversity was reduced in the METH users, showing significant differences in the microbiota and changes in oral metabolic pathways. The authors identified four key microbial taxa, i.e., *Peptostreptococcus*, *Gemella*, *Campylobacter*, and *Aggregatibacter*, involved in the toxicity and addiction mechanisms of METH by affecting the above metabolic pathways. Also, the content of tryptamine associated with neuropsychiatric disorders was gradually increased.

I have carefully reviewed the manuscript and found no major technical concerns. I believe that the authors have provided sufficient background, explained well the methodologies used in this study, presented the data with appropriate figures, and more importantly, withdrawn conclusions based on available data. I have some minor suggestions and a few typos listed here for the authors to consider if a revision is requested by the editor.

Title:

I would suggest the replacement of "Alterations" with "Variations"

Abstract:

Line 32: it is not clear what the "in this process" means, detoxification?

Line 40: the authors should indicate either "increase" or "decrease" of "with significant differences"

Line 47, delete "And,"

Line 48: "was gradually increased"

Introduction:

Line 86: I would suggest that the authors explicitly describe the goals of aims of this study somewhere in this paragraph.

Materials and Methods:

Line 105: delete "the"

Results:

The figures are well done, nice and informative.

Discussion:

I would suggest that the authors establish two subsections, microbiome and metabolome, to focus on the in-dept discussion of each of these two areas.

Reviewer #2 (Comments for the Author):

This study focused on the oral microbiology and metabonomics in methamphetamine abusers , provideing novel insights into exploring the toxic damage and addiction mechanisms underlying the METH addiction. Identification of fundamental variations in the microbiome and how the host will/can respond to this changing microbial burden is an important area for knowledge acquisition.

Generally, this report is well organized and informative. The methodology of the biology is sound and the statistical analysis and bioinformatics visualization of the data is good. However, I do have a number of minor to moderate concerns and one larger issue that adversely affects my enthusiasm for the report.

(1)The oral microbiota is influenced by many factors, such as the whole body and the local part of the mouth. The lifestyle of methamphetamine addicts is more complicated. Will there be differences in oral microbiota caused by oral health or oral hygiene problems? How to eliminate interference?

(2)The research object is the patients in the compulsory detoxification center, and the author's topic is "Alterations in the oral microbiome and metabolome of methamphetamine users ", so are the oral microorganisms the same during detoxification and non-detoxification? Has the author considered this problem?

(3)Line187 "In the discovery phase (194 METH users and 74 CTL)" , Line189"in the validation phase (METH 189 users and 31 CTL)" , Line190"In the discovery phase of metabolomics (119 METH users and 35 CTL)" , Line194"based on a total of 51 METH users and 15 CTL" , I would suggest including power calculation description to the above data in the manuscript.

This study focused on the oral microbiology and metabonomics in methamphetamine abusers, providing novel insights into exploring the toxic damage and addiction mechanisms underlying the METH addiction. Identification of fundamental variations in the microbiome and how the host will/can respond to this changing microbial burden is an important area for knowledge acquisition.

Generally, this report is well organized and informative. The methodology of the biology is sound and the statistical analysis and bioinformatics visualization of the data is good.

However, I do have a number of minor to moderate concerns and one larger issue that adversely affects my enthusiasm for the report.

(1)The oral microbiota is influenced by many factors, such as the whole body and the local part of the mouth. The lifestyle of methamphetamine addicts is more complicated. Will there be differences in oral microbiota caused by oral health or oral hygiene problems? How to eliminate interference?

(2)The research object is the patients in the compulsory detoxification center, and the author's topic is "Alterations in the oral microbiome and metabolome of methamphetamine users ", so are the oral microorganisms the same during detoxification and non-detoxification? Has the author considered this problem?

(3)Line187 “In the discovery phase (194 METH users and 74 CTL)” , Line189 “in the validation phase (METH 189 users and 31 CTL)” , Line190 “In the discovery phase of metabolomics (119 METH users and 35 CTL)” , Line194“based on a total of 51 METH users and 15 CTL” , I would suggest including power calculation description to the above data in the manuscript.

Dear Dr. Marshall:

Thank you very much for your kind E-mail message regarding our manuscript titled “Alterations in the oral microbiome and metabolome of methamphetamine users” (mSystems00991-23) submitted to your journal *mSystems* to be considered for publication. We appreciate very much your great efforts organizing the review of our manuscript. As requested, we have now revised our manuscript based on the comments and suggestions provided by the editors and reviewers. With this cover letter, we have also provided a point-to-point response to these comments and suggestions (see below). Simply, we have agreed on all of these comments and suggestions and revised our manuscript accordingly.

We have now submitted two versions of our revised manuscript, one with all of the changes marked using the Track Change Functionality of MicrosoftWord and the other a “clean” copy with the changes not marked.

Again, we appreciate very much the editors’ and reviewers’ constructive comments and suggestions. We hope that we have now revised our manuscript to your satisfaction. We now look forward to hearing from you for further instructions.

Best wishes,

Best regards!

Yours sincerely,

Yundong Sun,

Department of Microbiology, Key Laboratory for Experimental Teratology of Ministry of Education, School of Basic Medicine, Cheeloo College of Medicine, Shandong University, Jinan, Shandong 250021, China.

E-mail: syd@sdu.edu.cn

November 8, 2023

A point-by-point response to the reviewers' comments

We really appreciate the editor's professional evaluation on our manuscript. We have now revised our manuscript based on the comments and suggestions provided by the editors and reviewers.

To Reviewer 1:

We appreciate the reviewer's comments. The responses to your questions are as follows:

1. Title: I would suggest the replacement of "Alterations" with "Variations".

Response: Thank you very much for your suggestion. As suggested, we have changed "Alterations" to "Variations" in the title of our manuscript.

2. Abstract: Line 32: it is not clear what the "in this process" means, detoxification?

Response: We apologize for this confusion. The "this process" refers specifically to the process of drug addiction (lines 32).

3. Abstract: Line 40: the authors should indicate either "increase" or "decrease" of "with significant differences".

Response: As suggested, we have revised this sentence as follows: ... that compared to the CTL group, alpha diversity was reduced in the group of METH users, the relative abundance of *Peptostreptococcus* and *Gemella* were significantly increased, while the relative abundance of *Campylobacter* and *Aggregatibacter* were significantly decreased (lines 37-41).

4. Abstract: Line 47, delete "And,".

Response: We have now removed the "And," in line 47.

5. Abstract: Line 48: "was gradually increased".

Response: As suggested, we have now added the word "was" in line 48.

6. Introduction: Line 86: I would suggest that the authors explicitly describe the goals of aims of this study somewhere in this paragraph.

Response: As suggested, we have now revised the sentence as follows: ... to investigate the role played by the oral microbiome in METH addiction (lines 85-86).

7. Materials and Methods: Line 105: delete "the".

Response: We have removed the word "the" in line 108.

8. Discussion: I would suggest that the authors establish two subsections, microbiome and metabolome, to focus on the in-dept discussion of each of these two areas.

Response: As suggested, we have now established two subsections, i.e. "4.1 Variations in the oral microbiome" and "4.2 Variations in the saliva metabolome" in Discussion (lines 339 and 378).

To Reviewer 2:

We appreciate the reviewer's comments. The responses to your questions are as follows:

1. The oral microbiota is influenced by many factors, such as the whole body and the local part of the mouth. The lifestyle of methamphetamine addicts is more complicated. Will there be differences in oral microbiota caused by oral health or oral hygiene problems? How to eliminate interference?

Response: We agree with this reviewer on these related issues. Yes, indeed, the oral health conditions could affect differences in oral microbiota. To eliminate this interference, in this study, we examined the participants for oral hygiene according to the oral health criteria established by the World Health Organization: no pain, normal gum color, no bleeding, no caries, and clean teeth. We collected oral saliva only from participants who met these oral health criteria (lines 104-107).

2. The research object is the patients in the compulsory detoxification center, and the author's topic is "Alterations in the oral microbiome and metabolome of methamphetamine users", so are the oral microorganisms the same during detoxification and non-detoxification? Has the author considered this problem?

Response: We appreciate very much this reviewer for addressing this issue. It could be expected that with the extension of the duration of drug rehabilitation to ultimately achieving a successful rehabilitation, the oral microbiota of the drug users could be recovered to normal. However, given the difficulty of collecting oral microbiota from drug users, we consider the individuals with a shorter duration of drug rehabilitation or those who have not successfully completed rehabilitation active drug users. As indicated by this reviewer, indeed, if the drug rehabilitation is successful and the oral microbiota restores to normal, then new evidence would be provided by the oral microbiota to support the identification of drug abuse. This requires further research.

3. Line187 "In the discovery phase (194 METH users and 74 CTL)", Line189" in the validation phase (METH 189 users and 31 CTL)", Line190" In the discovery phase of metabolomics (119 METH users and 35 CTL)", Line194" based on a total of 51 METH users and 15 CTL" , I would suggest including power calculation description to the above data in the manuscript.

Response: We appreciate very much this reviewer for this suggestion of additional statistical analysis. Previous studies showed that the random forest model, by using the Bootstrap sampling method and both the strata and sampsize options in the randomForest function, the problem of imbalanced sample sizes could be avoided [1]. Additionally, research suggested that constructing multiple decision trees in the random forest model could reduce overfitting and improve the model's generalization ability [2]. In our study, we generated a random forest model using the Bootstrap sampling method, with the strata and sampsize options selected to overcome the issue of imbalanced samples. We also chose to include 500 decision trees to support the

adequacy of sample sizes in both the discovery and validation phases of our research (lines 176-179).

- [1] Bradley, A. P. The use of the area under the ROC curve in the evaluation of machine learning algorithms. *Pattern Recognition*. 1997; 30:1145-1159. doi.org/10.1016/S0031-3203(96)00142-2
- [2] Ho, T. K. Random decision forests. *Proceedings of 3rd International Conference on Document Analysis and Recognition*. Montreal. 1995; 1:278-282. doi: 10.1109/ICDAR.1995.598994.

Re: mSystems00991-23R1 (Variations in the oral microbiome and metabolome of methamphetamine users)

Dear Prof. Yundong Sun:

Your manuscript has been accepted, and I am forwarding it to the ASM production staff for publication. Your paper will first be checked to make sure all elements meet the technical requirements. ASM staff will contact you if anything needs to be revised before copyediting and production can begin. Otherwise, you will be notified when your proofs are ready to be viewed.

Featured Image Submissions: If you would like to submit a potential Featured Image, please email a file and a short legend to mSystems@asmusa.org. Please note that we can only consider images that (i) the authors created or own and (ii) have not been previously published. By submitting, you agree that the image can be used under the same terms as the published article. File requirements: square dimensions (4" x 4"), 300 dpi resolution, RGB colorspace, TIF file format.

Sincerely,
Christopher Marshall
Editor
mSystems